# Mammalian splicing divergence is shaped by drift, buffering in *trans*, and a scaling law

Xudong Zou[1,2,*] , Bernhard Schaefke[1,2,3,*] , Yisheng Li[2], Fujian Jia[2], Wei Sun[2], Guipeng Li[1,2,3], Weizheng Liang[2], Tristan Reif[4], Florian Heyd[4], Qingsong Gao[5], Shuye Tian[1,2], Yanping Li[2], Yisen Tang[2], Liang Fang[1,2,3] , Yuhui Hu[1,2] , Wei Chen[1,2,3]

**Alternative splicing is ubiquitous, but the mechanisms underlying its pattern of evolutionary divergence across mammalian tissues are still underexplored. Here, we investigated the *cis*-regulatory divergences and their relationship with tissue-dependent *trans*-regulation in multiple tissues of an F1 hybrid between two mouse species. Large splicing changes between tissues are highly conserved and likely reflect functional tissue-dependent regulation. In particular, micro-exons frequently exhibit this pattern with high inclusion levels in the brain. *Cis*-divergence of splicing appears to be largely non-adaptive. Although divergence is in general associated with higher densities of sequence variants in regulatory regions, events with high usage of the dominant isoform apparently tolerate more mutations, explaining why their exon sequences are highly conserved but their intronic splicing site flanking regions are not. Moreover, we demonstrate that non-adaptive mutations are often masked in tissues where accurate splicing likely is more important, and experimentally attribute such buffering effect to *trans*-regulatory splicing efficiency.**

## Introduction

Splicing is an important gene regulatory step in multicellular eukaryotes. Often the same pre-mRNA transcript can be spliced in various ways (Berget et al, 1977; Chow et al, 1977), leading to different combinations of exons in the mature transcript. This so-called

alternative splicing (AS) contributes to the variety of the cellular transcriptome/proteome as well as to the fine tuning of gene expression on the post-transcriptional level (Nilsen & Graveley, 2010; Kelemen et al, 2013).

Individual examples for important biological roles of AS have been extensively documented (Braunschweig et al, 2013; Kelemen et al, 2013; Raj & Blencowe, 2015; Baralle & Giudice, 2017), and several studies suggest that large changes in relative isoform abundance between tissues are functionally relevant (Wang et al, 2008; Reyes et al, 2013; Tapial et al, 2017). In particular, a special class of exon which is 3–30 nucleotides in length, called micro-exon, is abundantly regulated in the nervous system and plays a neural-specific function (Irimia et al, 2014; Li et al, 2015). Nevertheless, it has been proposed that to a certain degree AS between organs or cell types can be explained as splicing noise because only a small percentage of isoforms detected on the transcript level has been confirmed on the protein level (Abascal et al, 2015; Tress et al, 2017a, 2017b), and the role of AS and nonsense-mediated decay in regulating gene expression levels appears to be limited (Saudemont et al, 2017). Recent studies on alternative polyadenylation (Xu & Zhang, 2018), alternative transcriptional initiation (Xu et al, 2019), and stop-codon read-through (Li & Zhang, 2019) by the Zhang laboratory documented that generally lowly expressed genes exhibit more variation in these molecular phenotypes, supporting the "error hypothesis," that is, each gene has one optimal transcript isoform and alternative isoforms arise primarily from imprecise molecular processing rather than functional regulation.

A related question is the contribution of changes in AS to adaptive divergence between species. The neutral theory of molecular evolution states that most of the observed sequence

---

[1]Shenzhen Key Laboratory of Gene Regulation and Systems Biology, School of Life Sciences, Southern University of Science and Technology, Shenzhen, China [2]Department of Biology, School of Life Sciences, Southern University of Science and Technology, Shenzhen, China [3]Academy for Advanced Interdisciplinary Studies, Southern University of Science and Technology, Shenzhen, China [4]Institute for Biochemistry, Freie Universität Berlin, Berlin, Germany [5]Laboratory for Systems Biology and Functional Genomics, Berlin Institute for Medical Systems Biology, Max-Delbrück-Centrum für Molekulare Medizin, Berlin, Germany

Correspondence: chenw@sustech.edu.cn
Xudong Zou's present address is Institute of Systems and Physical Biology, Shenzhen Bay Laboratory, Shenzhen, China.
Bernhard Schaefke's present address is The Brain Cognition and Brain Disease Institute, Shenzhen Institute of Advanced Technology, Chinese Academy of Sciences; Shenzhen-Hong Kong Institute of Brain Science–Shenzhen Fundamental Research Institutions, Shenzhen, China.
Wei Sun's present address is Department of Pharmaceutical Chemistry and the Cardiovascular Research Institute, University of California San Francisco, San Francisco, CA, USA.
Qingsong Gao's present address is Department of Developmental Neurobiology, St. Jude Children's Research Hospital, Memphis, TN, USA.
*Xudong Zou and Bernhard Schaefke contributed equally to this work.

---

variation within and between species is selectively neutral (Kimura, 1979, 1989). Purifying selection and random drift are the major forces determining the fate of new mutations, with positive selection only playing a minor role. The neutral model has been widely accepted as the null hypothesis in studies of sequence evolution, and the approach has been extended to molecular phenotypes and cellular morphology (Ho et al, 2017; Xu & Zhang, 2018; Zhang, 2018). In comparison to tissue-dependent mRNA expression levels, AS patterns generally diverge more rapidly in vertebrates (Barbosa-Morais et al, 2012; Merkin et al, 2012; Schaefke et al, 2018), and numerous variants within humans have been found to affect splicing patterns across tissues (Amoah et al, 2021). Whereas evolutionary divergence of AS between closely related mammalian species can be largely attributed to *cis*-regulatory changes (Gao et al, 2015), the global significance of these changes and their mechanistic relationship with tissue-dependent *trans*-regulation remain undetermined.

To investigate tissue-dependent AS and its role in gene regulatory divergence in mammals, we use an F1 hybrid between two mouse species (C57BL/6J, a laboratory strain largely derived from the *Mus musculus domesticus* subspecies [Bonhomme et al, 2008], and SPRET/EiJ, a wild-derived inbred *Mus spretus* strain [Dejager et al, 2009]). Using high-throughput RNA-seq, we investigate the *cis*-regulatory component of AS divergence between the two strains across six organs and embryonic stem cells (ESC). We examine the effects of AS on coding sequences (CDS) and compare the splicing patterns of different genes with various gene expression levels and evolutionary conservation. Our results suggest that the general patterns of *cis*-regulatory divergence between the two species largely conform to the neutral hypothesis of molecular evolution. But in many cases non-adaptive *cis*-regulatory changes in AS only become visible in tissues where accurate splicing likely is of lesser importance, whereas they appear to be buffered in tissues where the gene is highly expressed. To test the role of the *trans*-regulatory splicing machinery in this buffering, we chemically perturbed the spliceosome in F1 hybrid fibroblasts. Indeed, the magnitude of allelic divergence increased, and the effects of previously buffered *cis*-regulatory mutations became unveiled after splicing perturbation.

In addition, we find that the magnitude of allelic splicing divergence is not only dependent on the number of variants in regulatory regions, but that a *cis*-regulatory mutation could lead to a change of greater magnitude for an event with intermediate inclusion levels than for one with either very high or very low inclusion levels. This finding is in accordance with a recent study describing a scaling law based on the kinetics of competition between splice sites (Baeza-Centurion et al, 2019), showing the large-scale effect of this mechanism on AS divergence.

## Results

### AS diversity within and across tissues reflects both molecular error and functional regulation

We obtained samples from five different organs (cerebral cortex, heart, lung, kidney, spleen) and ESC derived from F1 (C57BL/6J x

SPRET/EiJ) hybrid mice, and performed high-throughput RNA-seq for two biological replicates of each sample to quantify mRNA expression levels and AS pattern (see the Materials and Methods section; Fig 1A and Table S1). To cover more cell types, we included the liver data from the same animals obtained in our previous study (Gao et al, 2015).

After read mapping (see the Materials and Methods section; Table S1), we quantified the abundance of different mRNA isoforms, measured by percent spliced in (PSI) (see the Materials and Methods section), and estimated the quality of our data (Fig S1). PSI values derived from total reads (including unambiguously assigned allelic reads and common reads, see the Materials and Methods section) were used for examining splicing diversity within single tissues as well as across tissues. A splicing event was considered as expressed in a tissue, if a minimum of 20 informative reads (spliced-in + spliced-out) mapped to the event in both replicate samples (see the Materials and Methods section). Given that skipped Exon (SE) events constituted the AS category of the highest abundance (Table S2), we focused on SE events throughout our analyses. In total, 15,024 SE events were reliably detected in 6,660 protein-coding genes (Table S3). Among all SE events, 3,295 (21.9%) were expressed in only one tissue, 7,732 (51.5%) in two to six, and 3,997 (26.6%) in all the seven tissues (Fig S2A).

We then investigated the patterns of splicing diversity within individual tissues, in each of which we defined the percent dominant isoform (PDI) of each expressed event as the relative usage of the dominant isoform in this tissue. Among all tissues, cerebral cortex showed the highest splicing complexity, with 21.7% of the expressed genes being alternatively spliced and 5,325 (53.1%) expressed SE events having PDI values below 0.9 (Fig 1B and Table S4).

The error hypothesis states that only the dominant isoform of a gene is functional and predicts that its relative usage is higher for genes whose mis-splicing would have a large deleterious phenotypic impact. Genes with lower expression levels would therefore be expected to contain more events with lower PDI values than highly expressed genes. Overall, we observe only a negligible correlation between the minimum PDI (among all SE events of a gene) and mRNA expression level, but a weak one when only considering genes with PDI larger than 0.9 (Fig 1C and Table S5). These events also show a strong negative correlation between the PDI and the PSI variation between biological replicates (Table S5). This suggests that minor splicing isoforms might indeed predominantly reflect molecular error for events with a PDI above 0.9, whereas the pattern for the remaining events is less clear.

Previous studies suggest that large splicing changes between tissues might be functionally important (Wang et al, 2008; Reyes et al, 2013; Tapial et al, 2017). They could be highly regulated and affect genes under stronger selective constraints. Our results substantiate these findings. Events with a switch score (maximum |ΔPSI| between tissues, see the Materials and Methods section) > 0.5 frequently affect the coding region of a gene, and genes harboring these "Switch-Like" events have comparatively high expression levels, low dN/dS ratios, and high sequence conservation around the splice sites (Fig S2B–E; see the Materials and Methods section). Interestingly, we also found an enrichment of micro-exons in the "Switch-Like" category (Fig S2F), and consistent with previous

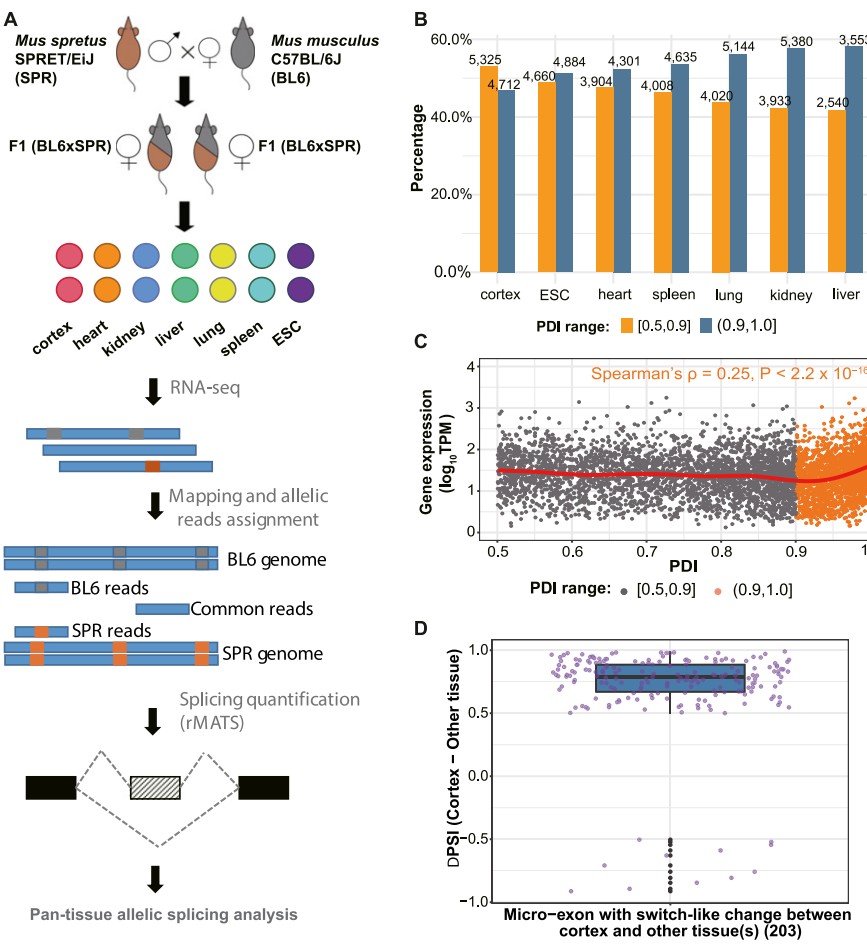

**Figure 1. Quantification of alternative splicing patterns and diversity within and across tissues.**
**(A)** Scheme of experimental design. **(B)** Proportion of high percent dominant isoform (PDI) (PDI > 0.9) and low PDI (PDI ≤ 0.9) SE events within each tissue. Numbers on the top of bars indicate the number of events in corresponding categories. **(C)** Correlation of PDI of SE event and gene expression of the corresponding gene in cerebral cortex. **(B)** Events are classified into two groups based on PDI values as in (B). The regression line (red) is fitted with a generalized additive model. See also Tables S1–S4. **(D)** 203 micro-exons exhibit Switch-Like changes between cerebral cortex and other tissue(s). The percent spliced in difference between cerebral cortex and other tissues (cortex–most different tissue) reveals higher inclusion levels in cerebral cortex than in other tissues. Box plot elements: center line, median; box limits, lower and upper quartiles; whiskers, lowest, and highest value within 1.5 IQR.

findings (Irimia et al, 2014; Li et al, 2015), these micro-exons often exhibit the highest inclusion levels in cerebral cortex (Fig 1D). Together, these results indicate that Switch-Like splicing differences between tissues are functionally important, especially in the brain, and result from tissue-specific regulation rather than noisy RNA processing. In contrast, many of those events with subtle changes across tissues or slight deviations from a PDI near one within a tissue may reflect fluctuations in the *trans*-regulatory splicing environment.

### The effects of *cis*-regulatory mutations on allelic splicing divergence are modulated by a scaling law

We then turned to *cis*-regulatory splicing differences between *M. musculus* and *M. spretus*, using only unambiguously assigned allelic reads (see the Materials and Methods section). Differences between the two alleles were considered significant if $|\Delta PSI_{AS}| \geq 0.1$ and the false discovery rate (FDR)-adjusted *P*-value < 0.05 (see the Materials and Methods section). As shown in Table S6, the AS pattern is more conserved between the two alleles in some tissues (e.g., cerebral cortex and kidney) than in others (e.g., heart and spleen). This is partially consistent with a previous study which examined AS patterns in nine tissues from four mammalian species

and chicken, in which the brain also exhibited higher conservation of splicing patterns (Merkin et al, 2012). As cerebral cortex exhibits the highest splicing complexity (Table S4), the high conservation of its splicing pattern indicates the potential functional relevance of different isoforms more prevalent in this tissue.

We then classified all events expressed in two or more tissues into three groups (Fig 2A): 1. "All-Divergent": the 298 events divergent in all expressing tissues; 2. "Some-Divergent": the 2,145 events divergent in some, but not all, of the expressing tissues; 3. "Non-Divergent": the 4,563 events not divergent in any of the expressing tissues.

*Cis*-divergence could result from sequence variants in the core splice sites and/or *cis*-regulatory elements in the flanking regions. To investigate the sequence features causing splicing divergence between the two alleles, we first compared the densities of sequence variants, that is, single nucleotide variants (SNVs) and insertions or deletions, in potential regulatory regions (the alternative exon and the 2 × 200 bp flanking intronic regions) between the three event categories (Fig 2B; see the Materials and Methods section). As shown in Fig 2B, All-Divergent events have higher density of variants than Some-Divergent events (mean density: 0.019 versus 0.016, one-sided Wilcoxon rank-sum test *P*-value = $6.18 \times 10^{-5}$), which in turn, have higher variant density than Non-Divergent events (mean density: 0.016 versus 0.015, one-sided

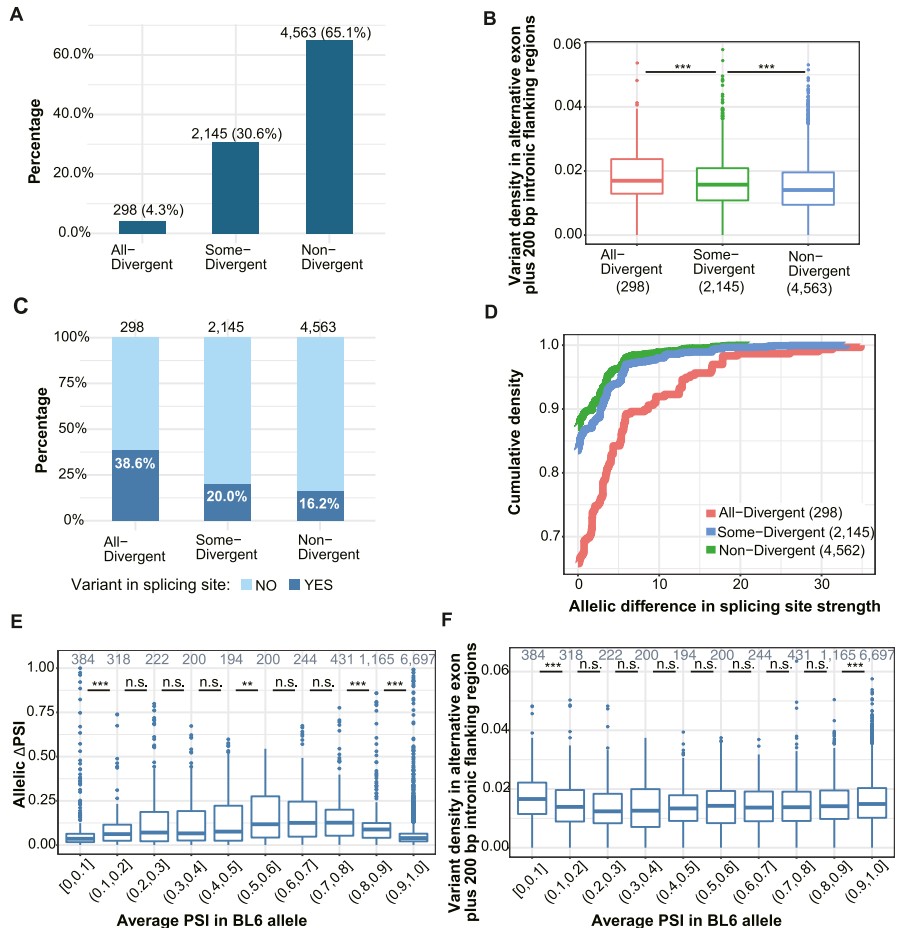

**Figure 2. *Cis*-regulatory sequence variants and scaling law affect allelic splicing divergence.**
**(A)** Classification of events expressed in two or more tissues into three groups based on allelic divergence across tissues. Numbers on the top of bars denote the count and percentage of events in corresponding groups. **(B)** Distribution of variant density in each divergence group. Variants in the alternative exon plus its intronic flanking regions (200 bp on both sides) were counted and divided by the total length of the region. Numbers in parentheses denote the numbers of events in corresponding groups (Two-sided Wilcoxon rank-sum test). **(C)** Percentage of SE events with (steel blue) or without (light sky blue) variants in splicing sites. Only the 5′ and 3′ splicing site of the alternative exon are considered. Numbers on the top of the bars indicate the total number of events within the corresponding groups (Fisher's exact test). **(D)** Cumulative distributions of the absolute difference between predicted scores for splicing site strengths of C57BL/6J (BL6) and SPRET/EiJ (SPR). Scores of the 5′ and 3′ splicing sites were summed up for each alternative exon. **(E)** Scaling law for allelic splicing divergence. The average BL6 percent spliced in (PSI) across tissues is treated as starting PSI, and the |ΔPSI| (y-axis) between SPR and BL6 is compared between different ranges of starting PSI (x-axis). Numbers on the top of boxes indicate the number of events in corresponding PSI ranges (one-sided Wilcoxon rank-sum test). **(F)** Distribution of variant density in alternative exon and its flanking introns (200 bp on both sides) for events in different PSI ranges. **(E)** The 10 bins of PSI are based on the average PSI of the BL6 allele across expressing tissues (starting allele in E). Boxplots show the Q1 to Q3 quartile values (the box limits), the median (the horizontal lines), and values within the 1.5 * IQR (the whiskers). n.s., not significant; *$P < 0.05$; **$P < 0.01$; ***$P < 0.001$.

Wilcoxon rank-sum test $P$-value = $2.93 \times 10^{-14}$). Furthermore, we also surveyed the 3′ and 5′ splicing sites flanking the alternative exon and compared the percentage of events with or without variants in these core splicing sites for the three categories. As shown in Fig 2C, the splicing sites of All-Divergent events are nearly double as likely to contain variants as those of Some-Divergent events (38.6% versus 20.0%, Fisher's exact test odds ratio = 2.5, $P$-value = $1.1 \times 10^{-11}$). In turn, significantly more Some-Divergent events contain variants in their splice sites than Non-Divergent events, but the difference is much less pronounced (20.0% versus 16.2%, Fisher's exact test odds ratio = 1.3, $P$-value = 0.0001; Fig 2C). Furthermore, for those containing variants in splicing sites, allelic differences in splicing strength of All-Divergent events were significantly higher than those of Some-Divergent and Non-Divergent events (two-sided K-S test $P$-value = $1.033 \times 10^{-8}$ and $P$-value = $4.725 \times 10^{-12}$, respectively; Fig 2D; see the Materials and Methods section). Consistent with this, the magnitude of allelic PSI differences for All-Divergent is also significantly higher than Some-Divergent events (Fig S3A–C). Among the All-Divergent events, those with a change in splicing site strength between the two strains had allelic differences of significantly higher magnitude than those without (Fig S3D–F). Together, these results indicate that large changes in core splicing elements are more likely to cause a larger difference in splicing, leading to significant divergence across all expressing tissues.

The magnitude of splicing divergence between two alleles does not only depend on the number of mutations affecting *cis*-regulatory elements, but also on the genetic background on which they occur. As a recently observed "scaling law" suggests, a single *cis*-regulatory mutation might lead to a change of greater magnitude for an event with intermediate inclusion levels (low PDI) than for an event with high PDI (Baeza-Centurion et al, 2019). To examine a possible scaling law affecting allelic splicing differences in our system, we asked whether divergent events with low PDI had larger differences between the two alleles than those with higher PDI (choosing the C57BL/6J allele arbitrarily as the reference). This was indeed the case (Fig 2E), even though events with PSI values above 0.9 or below 0.1 have slightly higher variant densities than events with intermediate PSI values (Fig 2F). Therefore, the results indicate that both *cis*-regulatory sequence variants and a scaling law affect allelic splicing divergence, and that events with high PDI appear to tolerate larger numbers of *cis*-regulatory mutations than those with low PDI.

## AS divergence is larger for genes under relaxed selective constraint

In general AS events with higher functional impact should be under stronger selective constraints and therefore more conserved

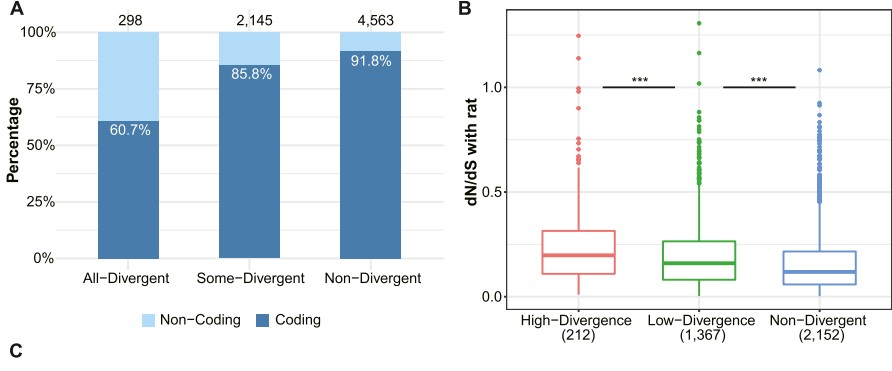

**Figure 3. Genes under relaxed selective constraints exhibit more divergence in alternative splicing.**
**(A)** Proportion of coding (steel blue) and non-coding (light sky blue) alternative exons in the three divergence groups. The percentage of coding alternative exons in each group is indicated within the corresponding bar, and numbers on the top of bars denote total numbers of events in the corresponding groups.
**(B)** Comparison of dN/dS ratios of genes belonging to different divergence groups (indicated by different colors) based on splicing divergence score score (see the Materials and Methods section). Numbers in parentheses indicate the number of genes within corresponding groups (one-sided Wilcoxon rank-sum test). **(C)** Comparison of average gene expression level among the three divergence groups. Numbers in parentheses indicate the number of genes within corresponding groups (one-sided Wilcoxon rank-sum test).

between species. When comparing events with different divergence patterns across tissues, we found that 90 percent of the Non-Divergent events affected coding regions of the transcript, whereas these were only about 85 percent for Some-Divergent events and 60 percent for All-Divergent ones (Fig 3A). Similarly, genes with faster protein sequence evolution or lower expression levels would be expected to exhibit larger splicing divergence than more conserved or highly expressed genes. To compare selective constraint and splicing divergence on the gene level, for each gene containing at least one SE event expressed in two or more tissues, we calculated a splicing divergence score (SDS; the average percentage of tissues with divergent splicing among expressing tissues of all events in the gene; see the Materials and Methods section). We divided all genes into three groups according to the SDS: (1) 2,373 Non-divergent genes (SDS = 0); (2) 1,509 genes with "Low Divergence" (SDS < 50); (3) 248 genes with "High Divergence" (SDS ≥ 50). It turned out that genes with high SDSs had higher average dN/dS ratios (mouse versus rat) than "Low Divergence" genes (Fig 3B, one-sided Wilcoxon rank-sum test $P$-value: 0.0003), followed by non-divergent genes (Fig 3B, one-sided Wilcoxon rank-sum test $P$-value: $1.14 \times 10^{-15}$). "High Divergence" genes also had lower average gene expression levels than "Low Divergence" genes (Fig 3C, one-sided Wilcoxon rank-sum test $P$-value: 0.013) which in turn had lower average gene expression levels than non-divergent genes (Fig 3C, one-sided Wilcoxon rank-sum test $P$-value: $5.56 \times 10^{-5}$). Similar patterns can also be observed on the event level (Fig S3G and H) and in individual tissues (Fig S3I and J).

Intuitively, the higher the gene is expressed, the more the junction reads it would generate, therefore likely resulting in more reliable PSI estimates. To check whether the observed lower allelic

divergence in highly expressed genes was not due to sampling bias, we created a down-sampled dataset in which the number of junction reads was equal for all the analyzed splicing events in either allele across all the samples (see the Materials and Methods section). As shown in Fig S4A–I, the PSI estimates were highly correlated between the original and down-sampled datasets, and more importantly, the lower allelic divergence in highly expressed genes could be observed also based on the down-sampled dataset (Fig S5A).

Interestingly, we also found that Switch-Like events are generally highly conserved between the two species (Fig S6 and Table S7), underscoring again their functional importance. Although most *cis*-divergent events between the two strains appear to reflect neutral or near-neutral drift, some might have a significant effect on the organismic phenotype. We found five candidate events affecting coding regions with substantial allelic divergence and high expression levels (see the Materials and Methods section), and therefore potential functional impact (Fig S7A–E). We were also able to identify several potential *cis*-regulatory changes which might cause the large divergence between the two alleles for two of these events (Fig S7A and B).

### Tissue-dependent patterns in AS divergence reveal buffering in *trans*

The biological effect of *cis*-regulatory differences between alleles depends on the *trans*-regulatory environment of each tissue. Interestingly, among the "Some-Divergent" events, gene expression levels were higher in non-divergent than in divergent tissues (Fig 4A). And for most events the magnitude of the allelic splicing difference showed a negative correlation with mRNA expression level across tissues (Fig 4B), indicating that mutations affecting

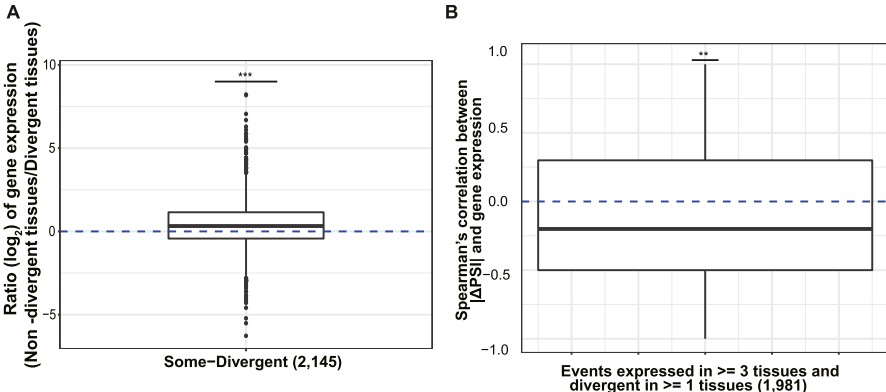

**Figure 4.   Negative correlation between gene expression and splicing divergence.**
**(A)** Comparison of average gene expression level of genes with "Some-Divergent" events in divergent tissues versus non-divergent tissues, the number in parentheses is the number of "Some-Divergent" events (One-sided Wilcoxon signed-rank test). **(B)** Distribution of Spearman's correlation coefficients between |ΔPSI| and gene expression across tissues for events expressed in ≥3 tissues and divergent in ≥1 tissue(s) (one-sided Wilcoxon signed-rank test). Box plot elements: center line, median; box limits, lower and upper quartiles; whiskers, lowest and highest value within 1.5 IQR. n.s., not significant; *$P$ < 0.05; **$P$ < 0.01; ***$P$ < 0.001.

splicing accuracy might be buffered in tissues where the gene is of more functional importance. Importantly, such phenomena could again be observed based on our down-sampled dataset, excluding the possibility that the observation is due to sampling bias (Fig S5B and C). Interestingly, we also found a negative correlation between gene expression level and ΔPSI between biological replicates across all the tissues, based on both original and down-sampled dataset (see the Materials and Methods section; Fig S8), indicating increased splicing noise for lowly expressed genes and further supporting our buffering hypothesis.

Patterns of allelic splicing divergence across tissues can be assigned to four different scenarios (Fig 5A): (1) both alleles exhibit tissue-dependent splicing, but the direction of change differs between alleles, (2) both alleles are spliced in a tissue-dependent manner with identical tissue-dependent isoform-preference, (3) tissue-dependent splicing occurs in only one allele whereas the other one shows no difference between tissues, (4) both alleles lack tissue-dependent splicing, but isoform usage is consistently divergent across tissues. To distinguish among these scenarios, for each event expressed in two or more tissues and divergent in at least one tissue, we compared the allelic $ΔPSI_T$ ($PSI_{T1} - PSI_{T2}$) in the tissue pair with the largest ΔΔPSI ($|ΔPSI_{Allelic\_T1} - ΔPSI_{Allelic\_T2}|$; see the Materials and Methods section; Fig 5B). We considered an event with $ΔPSI_T$ higher than 0.1 as differentially spliced between tissues. Of the 2,443 events, only 85 (3.5%) events exhibited opposite signs of $ΔPSI_T$ in the two tissues, fitting scenario 1. This shows that *cis*-divergence causing opposite allelic changes between tissues is rare, similar to the previously observed patterns for alternative polyadenylation (Li et al, 2020). Among the remaining events, 247 fit scenario 2 with parallel changes in isoform abundance between tissues and 456 fit scenario 4 with equal divergence across tissues, whereas the majority (1,655, 67.7%) fit scenario 3 with only one allele showing tissue-dependent splicing. The much higher frequency for scenario 3 in comparison to scenario 1 and 2 remained almost the same if different cutoffs (ranging from 0.1 to 0.2) were applied to determine differential splicing between tissues (see the Materials and Methods section; Table S8 and Fig S9).

We then compared the tissue-dependent regulation of the (more) variable allele between scenario 2 and scenario 3. Interestingly, as shown in Fig 5C, the maximum tissue difference is significantly higher in scenario 2, compared to scenario 3 (median value 0.45 versus 0.17, Wilcoxon rank-sum test, $P$ < 2.2 × 10$^{-16}$; Fig 5C).

This indicates that on one hand scenario 2 consists of more events with functional tissue-dependent regulation. On the other hand, for events in scenario 3, the allele with tissue-dependent splicing may be under noisy regulation. If this is true, we would expect that in scenario 3 the variable allele is less conserved. In contrast, if the allele with tissue-dependent splicing reflects functionally important regulation, that allele should show a more conserved pattern. To distinguish this, we took advantage of a published RNA-seq dataset (Thybert et al, 2018) for whole brain, liver, kidney, and heart from the closely related out-group species *Mus caroli* (Ryukyu mouse) and *Mus pahari* (Gairdner's shrewmouse). We considered all events expressed in at least two of the four tissues with out-group data and divergent in at least one of them, and classified them into the scenarios as described above (with the difference that we only consider the tissue pair with the largest ΔΔPSI among these four tissues and not all seven). In total, 826 events in the selected tissue pairs fit scenario 3 (Fig S10A). We compared their splicing patterns in the two tissues with their orthologous events in the respective tissues of the two out-group species (see the Materials and Methods section; Fig S10B). 170 events have orthologous events in at least one of the two out-groups, and indeed, among them the variable allele is less likely to exhibit the tissue-dependent splicing pattern conserved with the out-group species than the stable allele (23.5% versus 44.1%; Table S9). For these events with non-conserved variable allele, the observed tissue-dependent splicing might represent increased molecular error due to a lack of *trans*-regulatory buffering in the tissue where accurate isoform choice is less important.

When comparing micro-exons to larger exons, we found that micro-exons showing allelic divergence are enriched in scenario 2 (Fig 5D), and that the difference between scenario 2 and scenario 3 in the magnitude of tissue-dependent change is even higher for micro-exons (median value of $ΔPSI_T$ 0.62 versus 0.19, Wilcoxon rank-sum test, $P$ = 9.9 × 10$^{-13}$; Fig S10C) than for all exons as a whole (Fig 5C), again indicating the functional importance of tissue-dependent regulation in this exon class.

### Perturbation of the splicing machinery unmasks non-adaptive *cis*-regulatory changes

As mentioned above, for Some-Divergent events, in tissues where accurate splicing of a gene is important, both alleles often showed

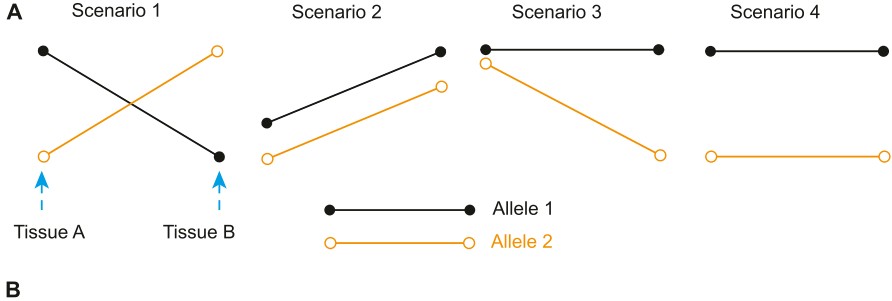

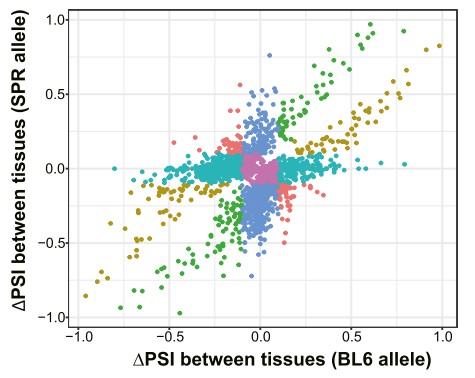

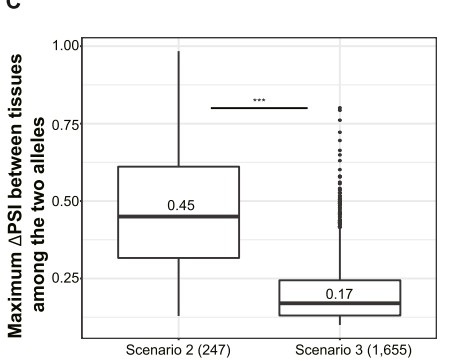

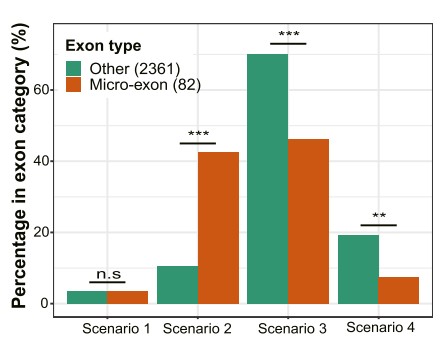

**Figure 5. Patterns of tissue-dependent allelic splicing divergence.**
**(A)** Scheme illustrating the four scenarios. Each segment indicates the pattern of splicing change of one allele between tissues and the two alleles are indicated with two different colors. **(B)** Allele-specific tissue-regulatory patterns. The x-axis presents the percent spliced in (PSI) difference of the C57BL/6J (BL6) allele between the two tissues with maximal difference in allelic ΔPSI, and the y-axis presents the corresponding information for the SPRET/EiJ (SPR) allele. **(C)** Scenario 2 holds higher PSI difference between tissues than Scenario 3. The maximum between-tissue PSI differences of the two alleles are compared among the four scenarios. Median value of the $\Delta PSI_T$ in each scenario is indicated in the box, and numbers in parentheses represent the numbers of events belonging to the corresponding scenarios. One-sided Wilcoxon rank-sum test was used to test for significance. **(D)** Micro-exons are enriched in Scenario 2 and less likely to be in Scenario 3 or 4. The bar plot shows the percentage of each scenario compared between micro-exons (dark orange) and other exons (forest green). Fisher's exact test was used to test for significance. n.s., not significant; *$P < 0.05$; **$P < 0.01$; ***$P < 0.001$.

conserved splicing patterns, even if mutations might have led to weaker *cis*-regulatory signals in one allele. We therefore hypothesized that in these "important" tissues, relevant *trans*-regulators, when expressed at high levels, could buffer against potentially deleterious *cis*-regulatory changes. In contrast, in tissues, where these splicing regulators are expressed at low levels, the phenotypic effects of *cis*-regulatory mutations would become visible.

To examine such buffering effect, instead of individual tissue-specific splicing regulators, which would only affect a limited number of events and therefore would not validate our buffering hypothesis in a statistically meaningful manner, we chose to perturb the general splicing machinery and then compared the allelic splicing pattern in our F1 system. More specifically, we treated a fibroblast cell line, previously derived from the same F1 hybrid mice (Gao et al, 2015), with pladienolide B, a macrocyclic lactone that inhibits mRNA splicing by selectively binding to the splicing factor 3B (SF3B) (Kotake et al, 2007; Yokoi et al, 2011), a key component of the spliceosomal U2 snRNP complex (Gozani et al,

1998). We expected that pladienolide B treatment of our F1 fibroblasts would result in a decrease in splicing accuracy and therefore unmask many of the previously unobserved *cis*-divergences.

We determined mRNA levels and splicing patterns similarly as those in F1 hybrid tissues (see the Materials and Methods section). Consistent with previous publications (Yoshimoto et al, 2017; Wu et al, 2018), pladienolide B treatment led to an increase in exon skipping and intron retention (Fig 6A). Given the focus of all our analyses on SE events, we considered the 3,583 SE events detected in both pladienolide B–treated and control (DMSO treated) samples. As shown in Fig 6B, both the percentage of allelic divergent events and the average magnitude of divergence increased dramatically after pladienolide B treatment (Fig 6B). Whereas 195 (5.4%) events were divergent only in the control sample, 670 (18.7%) were so in the treatment sample. This difference became even larger if we only considered highly divergent events (with |ΔPSI| ≥ 0.2): 295 (8.2%) events are highly divergent only in pladienolide B–treated cells, whereas 63 (1.8%) are highly divergent only in the control

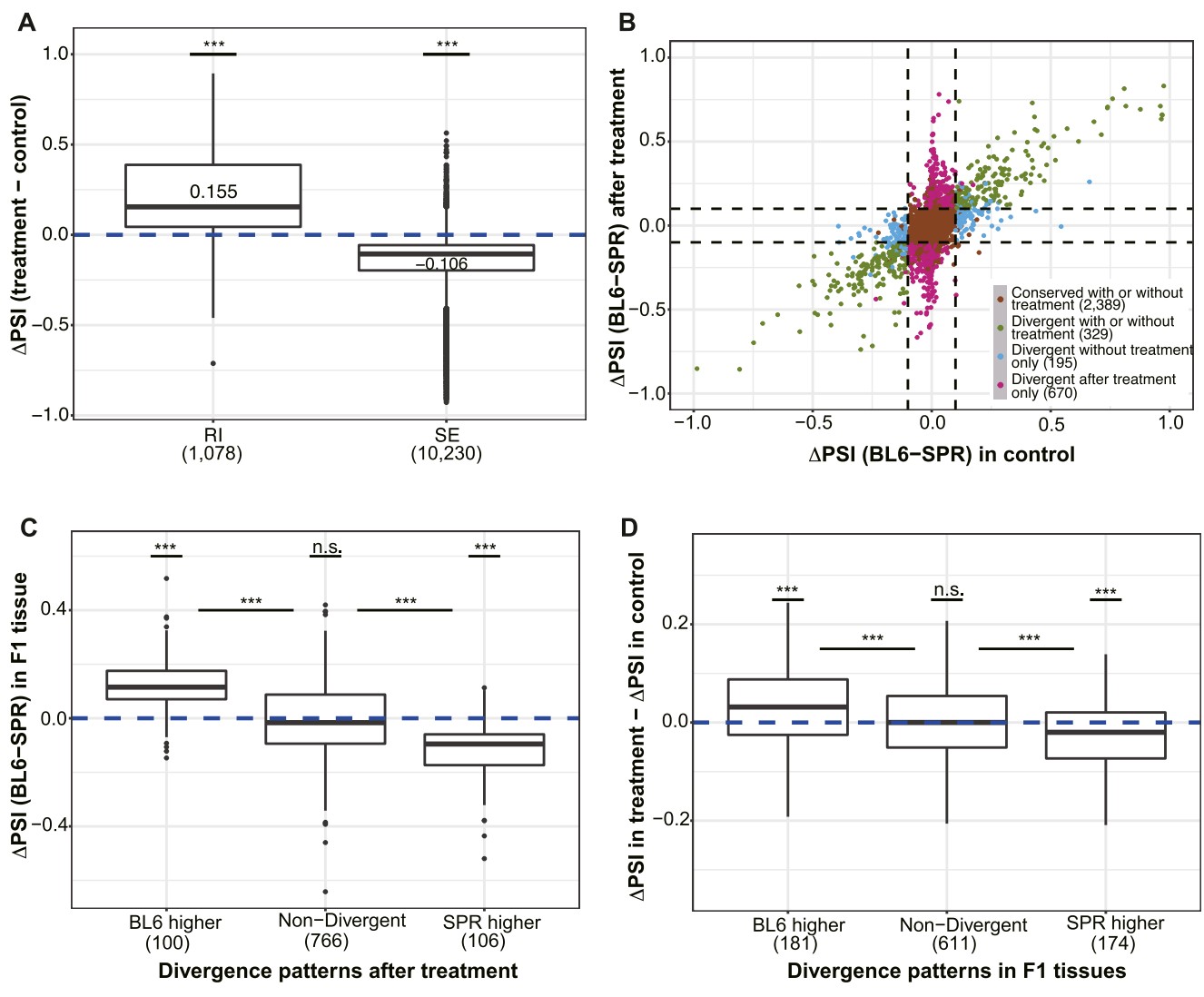

**Figure 6. Many previously buffered non-adaptive *cis*-regulatory changes are unmasked by perturbation of the splicing machinery.**
**(A)** Incidence of Retained Introns (RI) and Skipped Exons (SE) increased after pladienolide B (0.1 μM) treatment (one-sided Wilcoxon signed-rank test). **(B)** Increase in allelic splicing divergence after pladienolide B treatment. x-axis and y-axis are percent spliced in (PSI) divergence between the two alleles in the DMSO treated sample and the pladienolide B–treated sample, respectively. Events are classified into different groups based on PSI divergence in the two samples. The count of events in each group is indicated by numbers in parentheses. **(C)** Patterns of PSI divergence after pladienolide B treatment are consistent with those in other F1 tissues. Events were classified into different groups based on allelic splicing divergence after pladienolide B treatment, and the maximal ΔPSI between alleles in F1 tissues are compared for the above groups. **(D)** Divergent events in F1 tissues are also more likely divergent after pladienolide B treatment and with consistent direction. Events were classified into three groups based on the divergence pattern in F1 tissues, and the ΔPSI (BL6-SPR) values were compared between pladienolide B–treated samples and DMSO-treated samples. Outliers in each group are hidden and the y-axis has been limited to −0.35 to 0.35. One-sided Wilcoxon signed-rank test was used to test difference from zero for median values within each group, whereas one-sided Wilcoxon rank-sum test was used for comparing between groups. Box plot elements: center line, median; box limits, lower and upper quartiles; whiskers, lowest and highest value within 1.5 IQR. n.s., not significant; *$P < 0.05$; **$P < 0.01$; ***$P < 0.001$.

sample. We obtained similar results when using the fold change in splicing efficiency due to *cis*-mutations, that is, the additive effect "A" as described before (Baeza-Centurion et al, 2019; Baeza-Centurion et al, 2020), instead of ΔPSI as a measure of splicing divergence (see the Materials and Methods section; Fig S11).

We then examined the potentially unmasked events after pladienolide B treatment, that is, those non-divergent in control fibroblasts but divergent after treatment. Among 670 events, 206 were also expressed in at least one of the seven tissues we investigated. As shown in Fig 6C, these events showed significantly

higher allelic difference in the expressing F1 tissues than those that remain non-divergent after treatment. Importantly, for these events, the direction of divergence unmasked after treatment is largely consistent with that observed in the divergent tissues, indicating that the same *cis*-regulatory variants are contributing to the splicing differences observed after splicing perturbation. Furthermore, we looked from the other side and examined how different events categorized based on allelic data from the seven tissues responded to the pladienolide B treatment. As shown in Fig 6D, after perturbation, the divergent events defined based on the F1

tissue data showed higher magnitude of change in their allelic divergence than the non-divergent ones and again the allelic difference after treatment generally was in the same direction as the divergence observed in the F1 tissues. This further corroborated that under non-optimal splicing regulation, the *cis*-regulatory variants would result in higher allelic splicing divergence.

## Discussion

Our comprehensive analysis of allelic AS patterns in an interspecific mouse hybrid reveals that c*is*-regulatory divergence between the two strains mainly affects genes under relaxed selective constraints as predicted by the neutral theory of molecular evolution. Interestingly, for the same gene, expression levels also are generally higher in tissues in which its splicing pattern is conserved than in divergent tissues (Fig 3D). In 69.2% of these tissue-dependent divergent events, one allele exhibits no significant PSI differences between tissues ("stable" allele), but the other does ("variable" allele). The former more often represents the ancestral splicing pattern, based on the comparison with two out-group species. These findings support the hypothesis that in most cases the derived splicing pattern is the result of neutral or slightly deleterious *cis*-regulatory mutations, whose effects are masked in tissues where accurate splicing is essential. Therefore, it indicates that *trans*-regulatory factors are capable of buffering *cis*-regulatory mutations in tissues where they are highly abundant, ensuring accurate splicing of events even with "weaker" *cis*-regulation (e.g., the allele with reduced RBP binding affinity) and therefore acting as "buffers" or "phenotypic capacitors" (Rutherford & Lindquist, 1998). To test this hypothesis, we chose to chemically inhibit the splicing factor 3b (SF3B) with pladienolide B in our F1 hybrid fibroblasts because we assume that many auxiliary *trans*-regulators exert their effects through core components of the splicing machinery, for example, by increasing or decreasing the association of the splicing machinery to specific exon/intron junctions. Therefore, modest perturbation of core components might mimic that of auxiliary factors with tissue-dependent expression. As predicted, the percentage of divergent events and the average magnitude of splicing divergence between the two alleles both increased after treatment, suggesting that in many cases one allele with weak *cis*-regulatory elements is more susceptible to splicing error after this kind of perturbation and that larger error means larger deviation from an optimal splicing pattern common to both alleles. This is further supported by the finding that divergence after treatment increased in the same direction as found in the tissues where the event exhibits a *cis*-regulatory difference between the two alleles and that the change in allelic divergence is larger for the events showing higher allelic difference across tissues. The observed buffering could explain the genetic phenomenon where splicing mutations have incomplete penetrance or tissue-specific phenotypes (Scotti & Swanson, 2016), as their effects might depend on the genetic background of an organism, its physiological state, cell type, somatic mutations and the resulting *trans*-regulatory environment (Braunschweig et al, 2013). Whether the phenotypic capacitance observed in this study also leads to a significant potential for

adaptive evolution as suggested by theoretical considerations and simulations (Masel, 2006), or whether *cis*-regulatory variants, whose effects are masked under normal circumstances, lead to an increased risk for cancer or degenerative diseases (Park et al, 2018), deserves further investigation. It will also be important to understand how the expression of *trans* factors in different tissues is coordinated with that of their target genes and with epigenetic modifications to ensure accurate splicing of essential genes in a tissue.

Besides *trans*-regulatory buffering, we also find that events with very high PDI values can tolerate a larger number of *cis*-regulatory mutations without showing changes in splicing patterns. This scaling law has previously been experimentally tested and mathematically described (Baeza-Centurion et al, 2019) and might contribute to the robustness of splicing events with only one optimal isoform. Indeed, as shown in Fig S12A, the non-differential events with PDI values above 0.9 show high sequence conservation in the alternative exon, but paradoxically the lowest sequence conservation in the intronic flanking region. In contrast, for those Switch-Like events, where both isoforms are functionally important, *cis*-regulatory regions experience much stronger selective constraint and therefore exhibit higher conservation in both exon and intronic flanking regions (Figs S2E and S12A). It turned out that most non-differential events have a very high PDI, whereas the Switch-Like events are composed of those with lower PDI (Fig S12B). Therefore, whereas for both categories the exons consist of sequences of functional importance, and therefore are highly conserved, only Switch-Like events exhibit high conservation in intronic regions flanking splicing sites because the non-differential events with high PDI are comparatively robust to *cis*-regulatory perturbations. In the same manner, because they are also insensitive to the fluctuation of *trans*-regulatory environments, these events show hardly any changes in their splicing pattern across tissues. Therefore, for the events with only one optimal isoform, the robustness of their splicing pattern is largely hardwired.

Across tissues, on one hand, brain showed low allelic divergence (Fig S12C). In particular, only 7.5% of events expressed in brain and one other tissue exhibited divergence in brain, but no divergence in the other tissue. This percentage is the lowest among all the seven tissues, suggesting that the *trans*-regulatory splicing environment is most efficient in brain (Fig S12C). It also expressed the highest diversity of splicing isoforms, that is, more than half of the AS events were of PDI < 90%. Likely, both isoforms would be functional. However, given these events would be more sensitive to fluctuations of the *trans*-regulatory environment, the high splicing efficiency in brain is essential to ensure the stable/correct isoform compositions. This may explain why some splicing regulators are specific to or found in higher abundance in the nervous system (Raj & Blencowe, 2015; Vuong et al, 2016) and why neurological defects are often the most prominent symptoms of patients with mutations in splicing regulators (Singh & Cooper, 2012; Chen et al, 2019).

Although our results show that most of the divergence between closely related mammalian species fits the neutral model and that *cis*-regulatory mutations are frequently buffered in tissues where accurate splicing is important, we find five examples of splicing differences with a potential functional impact. Although it is possible that these divergent events are the result of slightly

deleterious mutations, especially for those found only in laboratory-derived mouse strains, some might also contribute to species- or strain-specific adaptations. It would be interesting to further examine in future studies whether the inclusion levels of any of these exons have undergone adaptive changes and contribute to lineage-specific phenotypes.

# Materials and Methods

## Samples and sequencing

Female F1 hybrid mice (female C57BL/6J x male SPRET/EiJ) used for tissue isolation were obtained as described before (Gao et al, 2013). All mice were kept in an air-conditioned, temperature-controlled conventional animal house and obtained standard chow and water ad *libitum*. Mice were euthanized at the age of 8 wk, and tissue samples from five organs (cerebral cortex, spleen, kidney, heart, and lung) were harvested as described before from two animals (Gao et al, 2015). ESC and fibroblasts were derived from the F1 hybrid mice. Two independent ESC clones were used in this study. For perturbing the splicing machinery, pladienolide B (#16538; Cayman) was prepared in DMSO at 0.1 mM, and used to treat fibroblasts at 0.1 μM for 4 h. Control fibroblasts were treated with 0.1% DMSO for 4 h. All animal husbandry and experiments were approved by the local ethical committee (VIB and Ghent University).

Total RNA was extracted using TriZOL reagent (Life Technologies) following the manufacturer's protocol. Then, stranded mRNA sequencing libraries were prepared with 500 ng total RNA. All samples were sequenced on a HiSeq 2000/2500 (Illumina) sequencer. Samples from heart, kidney and cerebral cortex were sequenced with paired-end reads of 101 bp in length. Samples from spleen, lung and ESC were sequenced with paired-end reads of 76 bp in length. The sequencing depth for each biological replicate was 240~260 million reads per sample, except for the two ESC samples, for which we obtained 175 million and 202 million reads, respectively (Table S1). Data for liver were obtained from Gao et al (2015).

## Public RNA-seq data for human (human body map 2.0)

Pan-tissue RNA-seq raw data of human were downloaded from the ArrayExpress database with accession number E-MTAB-513. The raw data were mapped using STAR (version 2.7.1a) with parameters "—runThread 40 –outSAMtype BAM SortedByCoordinate –alignEndsType EndToEnd."

## Reference genome and gene annotation

The reference *M. musculus* genome (mm10) and gene annotation of the C57BL/6J strain were downloaded from the Ensemble database (ftp://ftp.ensembl.org, version: GRCm38, release 74). SNVs and insertions/deletions (indels) between C57BL/6J and SPRET/EiJ were downloaded from the Mouse Genome Project (http://www.sanger.ac.uk/).

The vcf2diploid tool (version 0.2.6) in the AlleleSeq pipeline (Rozowsky et al, 2011) was used to construct the SPRET/EiJ genome by incorporating the SNVs and indels into the C57BL/6J genome. The

chain file between the two genomes was also reported as an output, which was further used with g2gtools to convert SPRET/EiJ coordinates to C57BL/6J coordinates.

## Mapping and allele-specific read assignment

To ensure that RNA-seq reads from all the samples have the same length, we trimmed 25 bp from the 3′ end of the 101 bp reads. We aligned the RNA-seq reads to the C57BL/6J reference genome and SPRET/EiJ genome separately with HISAT2 (Kim et al, 2015) (version 2.0.1) with parameters -p 12 -k 2 –reorder –no-softclip. Reads were assigned to the genome with less mapping edit distance. The reads with equal mapping distance to both genomes were designated as common reads. Genomic alignment coordinates of the reads that were assigned to SPRET/EiJ were then converted to the corresponding locations in the C57BL/6J reference genome using the g2gtools software (version 0.1.29).

Known imprinted genes deposited in the Geneimprint database (http://www.geneimprint.com/site/genes-by-species) and genes on the sex chromosomes or mitochondria were excluded from all analyses.

## AS analysis

The "replicate Multivariate Analysis of Transcript Splicing" (rMATS) (Shen et al, 2014) software was used with the default parameters for quantification and comparison of AS. The rMATS software counts the numbers of reads and the effective lengths of the inclusion isoform and the exclusion isoform (the number of unique isoform-specific read positions) to estimate the PSI value, representing the proportion of the inclusion isoform. For an event to be counted as "expressed" we required a minimum of 20 total reads (spliced-in + spliced-out) in each replicate sample when quantifying total splicing level (without distinguishing alleles) or in each allele when quantifying allelic splicing level. Events with PSI values in all expressing tissues either higher than 0.9 or lower than 0.1 in both replicates were not considered as alternatively spliced and therefore filtered out in the analyses based on total reads. For analyses of allelic splicing patterns, these events were only filtered if these conditions were also met by both alleles considered separately. We defined the PDI as PDI = |PSI − 0.5| + 0.5.

## Allelic AS analysis

Only the reads that could be unambiguously assigned to either genome were retained for estimating allele-specific AS in the F1 hybrid. Therefore, estimated allelic PSI and ΔPSI values might be inaccurate for events with lower SNP density. To avoid this potential error, using rMATs, we calculated and compared PSI values based on (1) the union of unambiguously assigned C57BL/6J reads and SPRET/EiJ reads only and (2) the union of C57BL/6J reads, SPRET/EiJ reads and common reads. Only the events for which the PSI values were consistent between the two datasets ($\Delta PSI_{consistency} \leq 0.1$ and FDR > 0.5) were retained for allelic analyses.

The allele-specific difference in inclusion levels ($\Delta PSI_{Allelic}$) for each event and each tissue was calculated as the difference between the average C57BL/6J inclusion level and the average

SPRET/EiJ inclusion level from the two biological repeats ($\Delta PSI_{Allelic} = PSI_{C57BL/6J} - PSI_{SPRET/EiJ}$). A hierarchical model to simultaneously account for sampling uncertainty in individual replicates and variability among replicates was used in rMATS as a measure of statistical significance for PSI differences. A combined threshold of FDR < 0.05 and average $|\Delta PSI_{Allelic}| \geq 0.1$ was used to define divergent events between two alleles in each tissue/cell line. We defined $\Delta\Delta PSI$ as the difference in allelic splicing divergence between two tissues ($\Delta\Delta PSI = |\Delta PSI_{Allelic\_T1} - \Delta PSI_{Allelic\_T2}|$).

### Definition of four scenarios of tissue-dependent allelic splicing divergence

Events expressed in two or more tissues and divergent in at least one tissue were classified into four different scenarios according to their allelic splicing patterns across tissues: (1) both alleles exhibit tissue-dependent splicing, but the direction of change differs between alleles, (2) both alleles are spliced in a tissue-dependent manner with identical tissue-dependent isoform-preference, (3) tissue-dependent splicing occurs in only one allele whereas the other one shows no difference between tissues, and (4) both alleles lack tissue-dependent splicing, but isoform usage is consistently divergent across tissues. To distinguish among these scenarios, we compared the allelic $\Delta PSI_T$ ($PSI_{T1} - PSI_{T2}$) in the tissue pair with the largest $\Delta\Delta PSI$ ($|\Delta PSI_{Allelic\_T1} - \Delta PSI_{Allelic\_T2}|$). If $\Delta PSI_T$ of both alleles is greater than the cutoff but with opposite direction, events are assigned to scenario 1; if $\Delta PSI_T$ of both alleles is greater than the cutoff and in the same direction, then events are assigned to scenario 2; if only $\Delta PSI_T$ of one allele is greater than the cutoff, these events are assigned to scenario 3; the remaining events are assigned to scenario 4, in which $\Delta PSI_T$ of both alleles is less than the cutoff. We repeated our classification with different cutoffs ranging from 0.1 to 0.2.

### Down-sampled dataset

To examine whether our estimates of allelic splicing divergence are biased because of sampling error, we down-sampled the assigned reads to the same level in either allele across all samples. Briefly, for each event of each allele in each sample, we randomly picked 20 reads from all junction reads supporting this splicing event, and calculated a PSI value measuring the inclusion level of the alternative exon based on these 20 reads. For total splicing level (without distinguishing between alleles), we also performed an analogous down-sampling analysis. For each event, we randomly picked 20 reads from all junction reads (including allelic reads and common reads) for each event, and calculated a PSI value based on these 20 reads.

### Calculation of switch score

To compare the AS patterns across tissues, we defined the "switch score" as the maximum pairwise absolute PSI difference ($|\Delta PSI_T|$) between expressing tissues, a measure also used in a previous study (Wang et al, 2008). Based on the switch score, events were classified into five groups: (1) "Non-Differential" with a switch score < 0.1; (2) "Low" with $0.1 \leq$ switch score < 0.2; (3) "Moderate-Low"

when the switch score was in the range of [0.2, 0.3); (4) "Moderate-High" for events with switch scores in the range of [0.3, 0.5); (5) "Switch-Like" in the case of events with switch scores $\geq 0.5$.

### Gene expression analysis

FeatureCounts (Liao et al, 2014) was first used to calculate the number of reads properly mapped to each gene in each tissue with parameters "-p -a $gtf -O -s 2 -t exon -g gene_id -T 36 –B –fraction." Fragments were counted once if they overlapped with multiple exons within the same gene. Fragments overlapping with more than one gene were assigned a fractional count to each overlapping gene (each overlapping gene receives a count of $1/y$ from a read, where $y$ is the total number of genes overlapping with the read). Transcripts per kilobase per million mapped reads (TPM) were used to quantify gene expression levels.

### Splicing site strength score analysis

For each splicing event, nine nucleotides from position −3 through six of the 5' splice site and 15 nucleotides from position −14 through 1 of the 3' splice site of alternative exons were extracted from the C57BL/6J and SPRET/EiJ genomes. These sequences were uploaded to the "Analyzer Splice Tool" server (http://host-ibis2.tau.ac.il/ssat/SpliceSiteFrame.htm) to calculate the splicing site strength score.

### Estimation of sequence conservation and variant densities

PhastCons scores of the Euarchontoglires clade (comprising rodents, rabbits, primates, and related species) were used to estimate sequence conservation. The pre-calculated PhastCons score data were obtained from the UCSC genome browser through the link: https://hgdownload.cse.ucsc.edu/goldenpath/mm10/phastCons60way/. Sequence variant density in selected regions (alternative exon plus 200 bp intronic flanking region on each side) was calculated based on SNVs and indels between C57BL/6J and SPRET/EiJ, one indel being counted once irrespective of its length.

### Potential effects of AS events and measurements of selective constraint

The coding sequence (CDS) annotation of each exon in the *M. musculus* mm10 genome was downloaded from the Ensembl database (release 75) via the BioMart interface. An event was defined to affect a coding region if the alternatively spliced region in this event overlaps with the coding sequence according to the CDS annotation.

To estimate selective constraints on the amino acid sequences of proteins, we used dN/dS ratios (ratio of the number of nonsynonymous substitutions per nonsynonymous site to the number of synonymous substitutions per synonymous site) (Miyata & Yasunaga, 1980) between the house mouse (*M. musculus*) and rat (*Rattus norvegicus*) downloaded from the ENSEMBL database (ensembl.org).

## Analysis of RNA-seq data from *M. caroli* and *M. pahari*

RNA-seq data of whole brain, heart, kidney, and liver produced in a previous study (Thybert et al, 2018) were downloaded from the European Nucleotide Archive (ENA, Study accession: PRJEB20980). Two biological replicates are available for each of the four tissues for *M. caroli* and one sample is available for each tissue for *M. pahari*.

The raw reads were aligned to the *M. caroli* reference genome (version 1.1) and *M. pahari* reference genome (version 1.1), respectively, by using HISAT2 (version 2.1.0) with similar parameters as used for aligning sequencing data from our F1 tissues (and with the additional parameters "--trim3 10 --trim5 5"). The rMATS software (version 4.0.2) was used for the detection and quantification of AS events in each dataset.

## Calculation of adjusted ΔPSI between replicates

The adjusted ΔPSI between replicates is defined for each event as the PSI difference between two replicates subtracting the PSI difference of each event between two mocked replicates. To generate two mocked replicates, for each event, we first combined reads supporting the inclusion isoform and reads supporting the exclusion isoform, keeping their label, and then randomly split them into two sets as two mocked replicates based on the original read numbers in replicate 1 and replicate 2. PSI values were calculated based on labeled reads in each mocked replicate using the rMATS model. This process was repeated 100 times and the $|\Delta PSI|_{mock}$ calculated as the average $|\Delta PSI|$ between mocked replicates. Finally, the adjusted $|\Delta PSI|$ can be calculated by subtracting $|\Delta PSI|_{mock}$ from $|\Delta PSI|$ between the two original replicates. The calculation processes were completed with in-house R scripts.

## Calculation of SDS

To measure splicing divergence at the gene level, we defined the SDS as the average percentage of tissues with divergent splicing pattern among all expressing tissues of all events in the gene. More specifically, for all genes containing $n$ events expressed in at least two tissues, we calculated the SDS as following:

$$SDS = \frac{\sum_i^n d_i/e_i}{n} \times 100,$$

where $d_i$ represents the number of tissues in which the AS event $i$ is divergent, and $e_i$ means the number of tissues in which the splicing event $i$ is expressed.

Genes are classified into three groups based on their SDS scores: the "Non-Divergent" group contains genes with SDS = 0, "Low Divergence" are genes with SDS < 50, and "High Divergence" are genes with SDS ≥ 50.

## Identification of orthologous events in out-group species

To identify orthologous events in the two out-group species (*M. caroli* and *M. pahari*) for selected events, the orthologous genes of the events were identified first according to the annotation in the gene transfer format files of the corresponding species downloaded from Ensembl. Then every exon in the orthologous gene was compared with the exons in the selected events, and only an exon with the same size and more than 90% sequence identity was treated as an orthologous exon. For an event to be considered orthologous, all the three relevant exons need to have orthologous exons in the same order as in C57BL/6J.

## Identification of ancestral and derived allelic splicing patterns

For events with allelic divergence and with ortholog(s) in at least one of the two out-group species (*M. caroli* and *M. pahari*), the C57BL/6J and the SPRET/EiJ allele were categorized as either derived or conserved in the following way (Fig S10B).

For each tissue, we calculated the average PSI value of the C57BL/6J and the SPRET/EiJ allele. If the PSI value of the orthologous event in the corresponding tissue of the out-group (cerebral cortex in our data was compared with whole brain in the out-group species) is within the range $PSI_{avg} \pm 0.05$ it was classified as "intermediate." Otherwise, it was classified as either C57BL/6J-like or SPRET/EiJ-like, depending on which allelic PSI value is closer. If only one out-group was available, being either C57BL/6J-like or SPRET/EiJ-like, or both out-group values were consistently classified as similar as to the same allele, this allele was designated as "conserved" and the other one as "derived." If the classification of the two out-groups was inconsistent (one C57BL/6J-like and one SPRET/EiJ-like) or both were classified as "intermediate," the conservation pattern was considered ambiguous in this tissue.

To determine the overall conservation pattern, we compared the tissue pair with the largest difference in ΔΔPSI among the four tissues with out-group data. If the same allele was classified as "conserved" (with ancestral splicing pattern) in both tissues, we used this classification for our analysis. If the two tissues showed opposite conservation patterns (in one tissue the C57BL/6J allele was classified as conserved and in the other tissue the SPRET/EiJ allele), the event was excluded. If both tissues were classified as "ambiguous," the overall conservation pattern was considered "ambiguous." In the remaining cases (one tissue with ambiguous and one with unambiguous conservation pattern), we determined the pattern based on the tissue with the larger magnitude of divergence between the C57BL/6J and the SPRET/EiJ allele.

## Identification of candidates of divergent splicing with potential functional relevance

Splicing events largely divergent between species may have functional impact on organismal phenotype if they occur on highly conserved exons and within highly expressed genes. To search for such kind of splicing events, we identified all candidate events with $|\Delta PSI| \geq 0.5$ in at least one tissue, and an expression level in the divergent tissue(s) no less than 30 (TPM). In addition, the maximal expression level of the gene in tissues with large divergence had to be equal to or higher than the average expression level of this gene across expressing tissues. Finally, alternative exon of the event is highly conserved (mean PhastCons score ≥ 0.8).

## Calculating the fold change of splicing efficiency between alleles

We also calculated fold change of splicing efficiency to measure splicing divergence (additive effects, "A") as used by Baeza-Centurion et al (2019) and Baeza-Centurion et al (2020). By taking one allele

(allele with the lower PSI) as starting allele, an "A" score was calculated according to the following equation:

$$A = \frac{PSI_s^2 - 100 \times PSI_s + \Delta PSI \times PSI_s - 100 \times \Delta PSI}{PSI_s(\Delta PSI + PSI_s - 100)},$$

where $PSI_s$ represents the starting PSI; $\Delta PSI$ denotes the PSI difference between the two alleles.

# Data Availability

Raw and transformed data are available from the Gene Expression Omnibus (accession GSE154727).

# Supplementary Information

# Acknowledgements

This work was supported by the Shenzhen Key Laboratory of Gene Regulation and Systems Biology (ZDSYS20200811144002008), Shenzhen Science and Technology Program (Grant No.: KQTD20180411143432337), Science and Technology Innovation Commission of the Shenzhen Municipal Government (Grant No.: JCYJ20180504165804015), Shenzhen-Hong Kong Institute of Brain Science-Shenzhen Fundamental Research Institutions (2019SHIBS0002), and National Natural Science Foundation of China (Grant Nos. 31861133013, 31970601, and 31771443). Bioinformatics analysis was supported by the Center for Computational Science and Engineering of Southern University of Science and Technology. We thank Dr. Jean Jaubert and Dr. Xavier Montagutelli from the Pasteur Institute for providing F1 hybrid mice, and Dr. Claude Libert and Dr. Tino Hochepied from the Vlaams Instituut voor Biotechnologie (VIB) for F1 hybrid embryonic stem cells.

## Author Contributions

X Zou: conceptualization, data curation, software, formal analysis, methodology, project administration, and writing—original draft, review, and editing.
B Schaefke: conceptualization, data curation, formal analysis, methodology, project administration, and writing—review and editing.
Y Li: data curation, formal analysis, and writing—review and editing.
F Jia: formal analysis.
W Sun: resources and data curation.
G Li: software and methodology.
W Liang: validation.
T Reif: formal analysis.
F Heyd: resources.
Q Gao: methodology.
S Tian: data curation and validation.
Y Li: data curation and validation.
Y Tang: data curation and validation.
L Fang: conceptualization and validation.
Y Hu: methodology and writing—review and editing.
W Chen: conceptualization, supervision, funding acquisition, investigation, methodology, project administration, and writing—review and editing.

## Conflict of Interest Statement

The authors declare that they have no conflict of interest.

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
