## [Reviewer comments · Life Science Alliance]

Life Science Alliance

Mammalian splicing divergence is shaped by drift, buffering in trans and a scaling law

Xudong Zou, Bernhard Schaefer, Yisheng Li, Fujian Jia, Wei Sun, Guipeng Li, Weizheng Liang, Tristan Reif, Florian Heyd, Qingsong Gao, Shuye Tian, Yanping Li, Yisen Tang, Liang Fang, Yuhui Hu, and Wei Chen

DOI: <https://doi.org/10.26508/lsa.202101333>

Corresponding author(s): *Wei Chen, Southern University of Science and Technology*

Review Timeline:

Submission Date:	2021-12-10
Editorial Decision:	2021-12-13
Revision Received:	2021-12-14
Editorial Decision:	2021-12-16
Revision Received:	2021-12-20
Accepted:	2021-12-20

Transaction Report:

Please note that the manuscript was previously reviewed at another journal and the reports were taken into account in the decision-making process at *Life Science Alliance*.

1st Review Round

Reviewer #1 Review

Report for Author:

This manuscript investigates the microevolution of tissue regulated alternative splicing in mammals using a mouse hybrid. They generated RNA-seq of various tissues and ESCs and analyze allele-specific differences in alternative splicing. In addition to the basic analyses with their tissue-specific RNA-seq, the authors further performed an experimental perturbation in fibroblasts, which I found quite a nice idea.

I found the analyses overall quite solid, the conclusions interesting and the manuscript enjoyable to read. I thus recommend publication. I have only two tiny comments:

- 1) The use of "higher" eukaryotes (first sentence of the Introduction), although common, should not be appropriate in a scientific context, especially in an evolutionary one. What do they refer to? Animals? Multicellular eukaryotes? Vertebrates?
- 2) I have missed an analysis of the potential cis-regulatory changes explaining the dramatic change in Apbb3 microexon inclusion. Do they detect any difference?

Reviewer #2 Review

Report for Author:

In their manuscript, Zou et al. investigate evolution of alternative splicing usage by measuring allele-specific splicing patterns in F1 hybrids from two diverged mouse species parents. Previous work from this lab measured allele specific splicing in these F1 hybrids in a single tissue to conclude that splicing divergence is largely driven by mutations acting in cis (Gao et al). This work extends the dataset across 6 organs. The finding that divergent splicing is generally non-adaptive but has identifiable signatures of selection is consistent with previous comparative splicing studies, and seems supported by their data. The most interesting potential contribution of this manuscript is the finding that genetic effects on splicing are often buffered in tissues where accurate splicing is important. I have some major concerns about the associated analysis and their interpretations (see below), which I believe severely limits the impact of this work. Finally, the authors attempt to prioritize five divergent splicing events that may explain phenotypic differences between the parental mouse strains. Though identification of splicing events that affect organismal phenotypes is an important and difficult problem, their analysis does not convincingly add anything in my view. This may be best reserved until follow up experiments can demonstrate the value that it adds, or at the very least, should come with more caution about the relevance of those five events.

Major specific points.

1. "But in many cases non-adaptive cis-regulatory changes in AS only become visible in tissues where accurate splicing likely is of lesser importance, while they appear to be buffered in tissues where the gene is highly expressed"

This comment relates to one of the paper's major conclusions as stated in the passage above (Fig. 3C-E), which show positive correlations between splicing differences (allelic PSI or divergence category) and gene expression. PSI estimates are often noisy due to low read counts over any particular splice junction. The 20 read count filter used by the authors (see methods) might mitigate this a bit, but especially when the minor isoform is very rare, randomness in sampling low read counts might still create very noisy estimates. Tissues in which a gene is higher expressed naturally will have more junction read counts, and possibly more reliable PSI estimates, resulting in smaller magnitude delta-PSI estimates between alleles/tissues for higher expressed splice events purely due to sampling. The authors should demonstrate this technical explanation is not driving the observations in Fig. 3C-E. In addition, the difference in gene expression level across the difference classes (High-divergence, Low-divergence, and Non-divergent) (Fig 3.C) is very small and thus it's difficult to convince myself that the authors' interpretation that splicing divergence are being buffered.

2. Related to interpretation of Fig. 4B, which shows the correlation of cross-tissue splicing changes for splice events with different allele, I am unsure of the contribution of noise. The text concludes that the majority of splice events with allelic effects have differing effect sizes between tissues (scenario 3), but all these scenarios exist on a spectrum, and the arbitrary classification cutoffs can be manipulated to classify events into different scenarios and tell a different story. Furthermore, problem I see with interpreting 4B in a meaningful way is that it is unclear how noisy the delta-PSI estimates are. For example, if the authors were to make the same plot using biological replicates to measure the same alleles across tissues and see a similar result at the presented thresholds for the 4 scenarios, I would be hesitant to interpret most of these scenario classifications in 4B as meaningful.

3. Some basic quality control figures should be included, and they would help address points 1 and 2: Correlation matrix of PSI values across all samples, including replicates. This would help give a sense of stability of PSI estimates. Also, the relationship between expression (read counts per event) and $|\text{delta-PSI}|$ between replicates (in the same tissue) would help address point 1.

4. Related to the idea that genetic effects are buffered in 'important' tissues:

We therefore hypothesized that in these "important" tissues, relevant trans-regulators, when expressed at high levels, could buffer against potentially deleterious cis-regulatory changes. In contrast, in tissues, where these splicing regulators are expressed at low levels, the phenotypic effects of cis-regulatory mutations would become visible.

To examine such buffering effect, instead of individual splicing regulators, we chose to perturb the general splicing machinery and then compared the allelic splicing pattern in our F1 system. More specifically, we treated a fibroblast cell line, previously derived from the same F1 hybrid mice (Gao, et al. 2015), with pladienolide B, a macrocyclic lactone that inhibits mRNA splicing by selectively binding to the splicing factor 3B (SF3B) (Kotake, et al. 2007; Yokoi, et al. 2011), a key component of the spliceosomal U2 snRNP complex (Gozani, et al. 1998). We expected that pladienolide B treatment of our F1 fibroblasts would result in a decrease in splicing accuracy and therefore unmask many of the previously unobserved cis-divergences.

• I am interested and also skeptical about the presented analysis to support this hypothesis. Firstly, in my view it is not a clear premise that inhibiting the core spliceosome machinery would reveal buffering effects of mutations that are dependent on tissue-specific auxiliary splicing components. But, in the author's defense, testing the inhibition of any single tissue-specific auxiliary components is unlikely to reveal many effects. Nonetheless, in presented PlaB spliceosome inhibition experiment, the abundance of points that tightly follow the vertical (Divergent after treatment only) in Fig5B (which shows changes in genetic effects sizes before and after PlaB treatment) is surprising to me. The authors interpret this as evidence of 'buffering', wherein many genetic effects are only uncovered when some of the buffering agents are inhibited (or when the core spliceosomal

machinery is partially inhibited in this case). My suspicion is these events on the vertical are enriched for intron retention events, where the intron is efficiently spliced in control (meaning not much room for large deltaPSI even if the fold-difference is large), but become retained after treatment. If so the allelic effects with and without treatment might be similar if measured in terms of fold-change rather than deltaPSI. In other words, the allelic effects on splicing still exist to a similar degree even when splicing is very efficient (ie changing from 1% to 2% PSI is still a two fold change, but only 1% deltaPSI), but upon PlaB treatment, you observe larger changes as expressed by deltaPSI (ie 10% to 20% change). The authors should redo the analysis in 5B after converting deltaPSI to fold-changes ('additive effects, A', as described by PMID 33112234) and see that there still lies an enrichment of points that are divergent only after treatment.

To some extent, my concern is a matter of semantics. The authors might argue that if an intron is efficiently spliced in terms of PSI, that regardless of the allelic fold change effect, the transcript is still efficiently spliced and final mRNA output is still 'buffered', even if the direct effects on splicing efficiency aren't 'buffered'. So authors should also more carefully communicate this section if they believe that additive effects are similar before and after PlaB treatment but that splicing effects are still buffered.

5. The highlighted events in fig7 don't add any value in my view. The section title "Highly divergent events potentially contribute to phenotypic evolution" is not supported. In this section, the authors highlight highly expressed, conserved exons that have large allele specific effects, and claim they potentially contribute to phenotypic evolution. From analysis of GWAS loci and eQTLs/sQTLs, we know that some small effect sizes in moderately expressed genes may have large organismal phenotypes. Or conversely, many large effect size genetic effects on splicing have no impact on organismal phenotypes. Are there any studies (GWAS) to support that these variants or loci have a phenotype on organism?

Minor points:

1. Fig1D and associated text:

"...these micro-exons often exhibit the highest inclusion levels in cerebral cortex"

I could be misinterpreting exactly what was done here... Wouldn't the above statement be true for any set of switch-like exons 'expressed' in any given tissue? The authors could show the same the exact analysis on other tissues for comparison. That is, find the set of switch like exons expressed in any given tissue, and plot the PSI between other tissues.

2. "To our knowledge this study presents the first comprehensive analysis of allele-specific AS across multiple tissues in a mammalian system."

There have been a number of studies that have compared genetic effects of splicing in various mammalian tissues. For example authors could cite some of the allele specific splicing papers that have worked on multiple tissues in humans from GTEx project (eg PMID: 33452016)

2nd Review Round

Reviewer #2 Review

Report for Author:

Zou et al responded with a generally nice point-by-point to my original review. Overall, I feel the manuscript has improved and is biologically interesting. However, a couple of my original concerns remain: More specifically, I worry that the most interesting conclusion (in my view), that splicing is buffered in "important" tissues, relies on an analysis that may be subtly but importantly flawed (see below for more detail).

Here I'll list my original points listed in their point-by-point response, and cross off the ones I feel are adequately addressed, before further discussing the ones that I feel are not well addressed:

Point1

Point2

Point3

Point4A

Point4B

Point4C

Point1

The NEGATIVE correlation (apologies for a typo in my original review where I wrote "positive" correlation) between allelic splicing differences and gene expression (Figure3) could be due to statistical sampling rather than some biological buffering of splicing in "important" tissues. That is, lowly expressed genes may have statistically noisier splicing estimates thereby upwardly biasing the change-in-splicing estimates (Δ PSI). The authors addressed this by down-sampling reads for all splice events (in both highly and lowly expressed genes) to equal levels (FigureR2). More details in the section below with R code embedded:

Simulating subsampling reads to calculate Delta PSI

Δ PSI is the change in percent spliced in between tissue. It is estimated by counting the fraction of RNA-seq reads supporting an exon as spliced in versus skipped. Sometimes, for any given junction there are as few as 20 or less splicing-informative reads, so PSI estimates for lowly expressed genes or exons can be noisy just due to random sampling, and as a result Δ PSI estimates

will also be noisy and the absolute value of Δ PSI estimate may be bigger for lowly expressed genes. So it is important to verify that any correlations between Δ PSI and expression aren't due to statistical counting artefacts. The authors attempted to address this concern: they randomly down-sampled spliced reads for estimating PSI down to 20 reads, for both low and high expressed genes. But I am skeptical they did this properly... In FigureR2 they present the correlation between Δ PSI before and after this downsampling process. The correlation looks too good to be true for 20 reads, so I slightly suspect a bug in their code. To better understand how the author's sub-sampling should in theory affect Δ PSI estimates, I used R's random sample generating functions to simulate correlation in Δ PSI estimates based on 20 reads per event, and compare it to a simulated ground truth for Δ PSI.

First I simulated some PSI estimates for 1000 exons in TissueA and TissueB, and plot a histogram of the simulated ground truth for Δ PSI across the 1000 exons

```
# a vector of 1000 PSI values, sampled from a uniform distribution
# These will represent the true biological value of PSI in tissue A
PSI.Tissue.A <- runif(1000)
```

```
# Another random sample, to simulate PSI in tissue B
PSI.Tissue.B <- runif(1000)
```

```
# Delta PSI is just the difference between PSI in A and PSI in B.
Delta.PSI.GroundTruth <- PSI.Tissue.A - PSI.Tissue.B
```

```
# histogram of delta PSI
hist(Delta.PSI.GroundTruth)
```

In figureR2 the authors subsampled 20 reads for each splice event and plot the correlation between Δ PSI before sampling and the Δ PSI based on only sampling 20 reads for each event in each tissue. For a single event in a single tissue, assuming that the original dataset had many many more reads per event (eg, $N \gg n$), the downsampled spliced reads to calculate PSI theoretically follows a binomial distribution:

```
SplicedReads=Binomial(size,p)
```

where size=20 trials, p=PSI, and

```
PSI_Observed=SplicedRead/20
```

Here, let's simulate observations of PSI based on 20 reads:

```
NumReadsSampledPerEvent <- 20
```

```
# A Binomial sample of size=20 reads to represent the number of reads spliced in. One observation for each of 1000
probabilities (True PSI values in tissue A).
```

```
SpicedReads.Tissue.A <- rbinom(n=1000, size=NumReadsSampledPerEvent, p=PSI.Tissue.A)
```

```
# Divide by the total number of reads to get PSI.
```

```
Observed.PSI.Tissue.A <- SpicedReads.Tissue.A/NumReadsSampledPerEvent
```

```
# Repeat for Tissue B
```

```
SpicedReads.Tissue.B <- rbinom(n=1000, size=NumReadsSampledPerEvent, p=PSI.Tissue.B)
```

```
Observed.PSI.Tissue.B <- SpicedReads.Tissue.B/NumReadsSampledPerEvent
```

```
# Calculate observed delta PSI
```

```
Observed.Delta.PSI <- Observed.PSI.Tissue.A - Observed.PSI.Tissue.B
```

```
plot(Delta.PSI.GroundTruth , Observed.Delta.PSI)
```

The correlation is good, but clearly not as good shown in FigureR2.

Upon further thought, I realize I can't actually conclude much from this because I compared sampling 20 reads to ground truth, analogous to comparing draws from an urn with replacement or when the number of balls in the urn is much much larger than 20 ($N \gg n$). In contrast, FigureR2 (as I understand it) represents sampling without replacement from the original dataset, analogous to an urn where the number of balls in the urn is not necessarily much larger than 20. So a better representation of what I should expect in FigureR2 would be if I simulated sampling with replacement from the full dataset. Let's simulate that process now, where every splice event has 40 reads in the original dataset, then 20 reads sampled without replacement from the original dataset. Sampling with replacement is modeled with hypergeometric distribution. Analogous to the number of white balls drawn after k draws without replacement from an urn with m white balls (SpicedReads) and n black balls (40-SpicedReads). So first let's redo the binomial sampling with 40 reads per event per tissue to represent the full dataset before downsampling, then simulate the downsampling to 20 reads per event per tissue:

```
NumReadsSampledPerEventInFullDataset <- 40
```

```
NumReadsSubSampledFromFullDataset <- 20
```

```
##### CODE basically copied from the block above to represent the sampling of RNA molecules to generate the full dataset
```

```
SpicedReads.Tissue.A <- rbinom(n=1000, size=NumReadsSampledPerEventInFullDataset, p=PSI.Tissue.A)
Observed.PSI.Tissue.A <- SpicedReads.Tissue.A/NumReadsSampledPerEventInFullDataset
```

```
SpicedReads.Tissue.B <- rbinom(n=1000, size=NumReadsSampledPerEventInFullDataset, p=PSI.Tissue.B)
Observed.PSI.Tissue.B <- SpicedReads.Tissue.B/NumReadsSampledPerEventInFullDataset
```

```
Observed.Delta.PSI.FullDataset <- Observed.PSI.Tissue.A - Observed.PSI.Tissue.B
#####
```

```
### Now let's simulate sampling with replacement from the full dataset
```

```
# For tissue A
```

```
SpicedReads.Tissue.A.AfterDownsampling <-
rhyper(nn = 1000,
m = SpicedReads.Tissue.A,
n = NumReadsSampledPerEventInFullDataset-SpicedReads.Tissue.A,
k = NumReadsSubSampledFromFullDataset)
Observed.PSI.Tissue.A.AfterDownsampling <-
SpicedReads.Tissue.A.AfterDownsampling/NumReadsSubSampledFromFullDataset
```

```
# And for tissue B
```

```
SpicedReads.Tissue.B.AfterDownsampling <-
rhyper(nn = 1000,
m = SpicedReads.Tissue.B,
n = NumReadsSampledPerEventInFullDataset-SpicedReads.Tissue.B,
k = NumReadsSubSampledFromFullDataset)
Observed.PSI.Tissue.B.AfterDownsampling <-
SpicedReads.Tissue.B.AfterDownsampling/NumReadsSubSampledFromFullDataset
```

```
# Calculate DeltaPSI
```

```
Observed.Delta.PSI.SubsampledDataset <-
Observed.PSI.Tissue.A.AfterDownsampling - Observed.PSI.Tissue.B.AfterDownsampling
```

```
### Now plot the DeltaPSI in the full dataset versus the subsampled dataset
plot(Observed.Delta.PSI.FullDataset, Observed.Delta.PSI.SubsampledDataset)
```

This correlation is not nearly as good as FigureR2. Maybe I should be more lenient and assume even fewer than 40 reads per junction are in the full dataset... If I now assume only 25 reads per junction in the full dataset, the sub-sampled dataset should better represent the full dataset, so the correlation may improve to something closer to FigureR2:

```
### This code block is identical to the above code block except that I changed this variable from 40 to 25
```

```
NumReadsSampledPerEventInFullDataset <- 25
```

```
###
```

```
NumReadsSubSampledFromFullDataset <- 20
```

```
SpicedReads.Tissue.A <- rbinom(n=1000, size=NumReadsSampledPerEventInFullDataset, p=PSI.Tissue.A)
Observed.PSI.Tissue.A <- SpicedReads.Tissue.A/NumReadsSampledPerEventInFullDataset
```

```
SpicedReads.Tissue.B <- rbinom(n=1000, size=NumReadsSampledPerEventInFullDataset, p=PSI.Tissue.B)
Observed.PSI.Tissue.B <- SpicedReads.Tissue.B/NumReadsSampledPerEventInFullDataset
```

```
Observed.Delta.PSI.FullDataset <- Observed.PSI.Tissue.A - Observed.PSI.Tissue.B
```

```
SpicedReads.Tissue.A.AfterDownsampling <-
rhyper(nn = 1000,
m = SpicedReads.Tissue.A,
n = NumReadsSampledPerEventInFullDataset-SpicedReads.Tissue.A,
k = NumReadsSubSampledFromFullDataset)
Observed.PSI.Tissue.A.AfterDownsampling <-
SpicedReads.Tissue.A.AfterDownsampling/NumReadsSubSampledFromFullDataset
```

```
SpicedReads.Tissue.B.AfterDownsampling <-
rhyper(nn = 1000,
m = SpicedReads.Tissue.B,
n = NumReadsSampledPerEventInFullDataset-SpicedReads.Tissue.B,
```

```
k = NumReadsSubSampledFromFullDataset)
Observed.PSI.Tissue.B.AfterDownsampling <-
SpicedReads.Tissue.B.AfterDownsampling/NumReadsSubSampledFromFullDataset
```

```
Observed.Delta.PSI.SubsampledDataset <-
Observed.PSI.Tissue.A.AfterDownsampling - Observed.PSI.Tissue.B.AfterDownsampling
```

```
### Now plot the DeltaPSI in the full dataset versus the subsampled dataset
plot(Observed.Delta.PSI.FullDataset, Observed.Delta.PSI.SubsampledDataset)
```

The correlation is better but still is not as good as FigureR2, and I felt I was being generous by assuming as little as 25 reads per junction in the original dataset. I therefore suspect a flaw in the author's subsampling procedure such that it does not convincingly address my original concern about some of the author's biologically interesting findings being potentially driven by natural statistical sampling.

Point4B

I appreciate the analysis that uses additive effect sizes rather than the DeltaPSI. The number of points on the vertical in 6B is astounding to me, and I caution interpreting it as the authors do: that there is an strong asymmetric enrichment of events that have larger allelic differences in splicing after spliceosome inhibition. It is just unique to see such an asymmetric effect, so (regardless of whether intron retention events were excluded) I would like to know if there is any technical source of this asymmetric bias. Specifically, are the exons chosen to be analyzed (and more specifically the ones on the vertical) symmetrical with respect to their prevalence in treated and untreated conditions? Furthermore, rather than just showing the additive effect sizes genome-wide increase in treated (FigR7), can you show that this is really what is happening to the highlighted events (on the vertical) in Figure6B. In other words, if you plot 6B with additive effects instead of deltaPSI, are there still lots of events on the vertical? If those quality controls check out, I will be more reassured that the author's results are not due to some technical artefact.

General comments:

In various places, the authors bin/categorize continuous values and correlating the bins/categories with continuous values. There is nothing problematic about this that should prevent publication in my view. But, simply correlating the continuous data with continuous data when possible is preferable to me. It is usually more sensitive statistically (higher resolution), and often simpler to interpret as it does not require arbitrary binning thresholds and category definitions. Of course, sometimes categorization can be useful for more simply communicating complex relationships, but sometimes it does the opposite: making simple relationships more complex to communicate. For example, rather than categorizing ("Some divergent", "All-divergent", "Non-divergent"), one could have just left it as the higher resolution categories ("Number of tissue pairs divergent").

December 13, 2021

Re: Life Science Alliance manuscript #LSA-2021-01333-T

Prof Wei Chen
Southern University of Science and Technology
Department of Biology
1088 Xueyuan Avenue
Shenzhen, Guangdong 518005
China

Dear Dr. Chen,

Thank you for submitting your manuscript entitled "Mammalian splicing divergence is shaped by drift, buffering in trans and a scaling law" to Life Science Alliance. We invite you to submit a revised manuscript and point-by-point response to address Reviewer 2's comments.

Thank you for this interesting contribution to Life Science Alliance. We are looking forward to receiving your revised manuscript.

Sincerely,

B. MANUSCRIPT ORGANIZATION AND FORMATTING:

To the Editor of
Life Science Alliance

Prof. Wei Chen
Department of Biology
Southern University of Science and
Technology

Tel: +86 (0) 755 88018449

chenw@sustech.edu.cn

December 14, 2021

Dear Dr. Sawey,

Thank you very much for your editorial efforts and thank the reviewer #2 for her/his constructive suggestions. In the revised manuscript, we have addressed all the concerns. Below is our point-by-point responses to the reviewer's comments. All changes in the revised manuscript have been marked in yellow. I hope that you will find our revised manuscript suitable for publication at LSA, am looking forward to hearing from you.

With best wishes,

wei

Yours sincerely,

Wei Chen, Chair Professor
Laboratory of Functional Genomics and Systems Biology
Department of Biology
Southern University of Science and Technology

Point 1:

The NEGATIVE correlation (apologies for a typo in my original review where I wrote “positive” correlation) between allelic splicing differences and gene expression (Figure3) could be due to statistical sampling rather than some biological buffering of splicing in “important” tissues. That is, lowly expressed genes may have statistically noisier splicing estimates thereby upwardly biasing the change-in-splicing estimates (ΔPSI). The authors addressed this by down-sampling reads for all splice events (in both highly and lowly expressed genes) to equal levels (FigureR2). More details in the section below with R code embedded:

Simulating subsampling reads to calculate Delta PSI

ΔPSI is the change in percent spliced in between tissue. It is estimated by counting the fraction of RNA-seq reads supporting an exon as spliced in versus skipped. Sometimes, for any given junction there are as few as 20 or less splicing-informative reads, so PSI estimates for lowly expressed genes or exons can be noisy just due to random sampling, and as a result ΔPSI estimates will also be noisy and the absolute value of ΔPSI estimate may be bigger for lowly expressed genes. So it is important to verify that any correlations between ΔPSI and expression aren't due to statistical counting artefacts. The authors attempted to address this concern: they randomly down-sampled spliced reads for estimating PSI down to 20 reads, for both low and high expressed genes. But I am skeptical they did this properly... In FigureR2 they present the correlation between ΔPSI before and after this downsampling process. The correlation looks too good to be true for 20 reads, so I slightly suspect a bug in their code.

Answer:

We thank the reviewer for the constructive criticism and for drawing our attention to a possible bug in our code. Indeed, in the down-sampling process, instead of using a single down-sampled dataset for analysis and then repeating the process for 100 times to test for robustness, we have mistakenly performed analysis using the average of the 100 down-sampling datasets. As expected and also suspected by the reviewer, this gave an almost perfect correlation between the down-sampled dataset and the original dataset. Now, we corrected the code and plotted the previous Figure R2 (here as Figure R1) with one of the down-sampled data sets as an example. We repeated the down-sampling process for 100 times, as originally intended, to obtain the mean and the 95% confidence interval

for the proportion of consistently classified events (Fig. R1H) and the correlations between allelic Δ PSI in the original and the downsampled data set (and Fig. R1I). As shown in the updated Figure R1, the correlation between the PSI differences in the original dataset and the down-sampled dataset is still good (Pearson's R greater than 0.9). To check whether the negative correlation between PSI divergence and gene expression still held, we updated Figures EV5 and Figure EV8 (new Fig S5 and Fig S8) with the corrected down-sampled data set. As shown in Fig S5 and Fig S8, we can still observe a statistically significant negative correlation, although as expected, the correlation became slightly weaker for the down-sampled dataset.

Figure R1. Comparing splicing divergence patterns between down-sampled dataset and original dataset. (A-G) We down-sampled junction reads of each event in either allele across all

samples to the same level of 20 reads, and determined divergent and non-divergent splicing based on PSI difference (divergent: $\Delta\text{PSI} \geq 0.1$). The X-axis and Y-axis denote PSI difference before down-sampling and after down-sampling, respectively. The colors distinguish “Divergent” (red) and “Non-Divergent” (light blue) in the original data set. The proportions of consistent classification of “Divergent” events and “Non-Divergent” events between down-sampled and original datasets have been labeled near the corresponding points. **(H)** The mean and 95% confidence interval (calculated by repeating the down-sampling process for 100 times) of proportions of consistent classification of “Divergent” events (red) and “Non-Divergent” events (blue) between down-sampled and original datasets. **(I)** The mean and 95% confidence interval (calculated by repeating the down-sampling process for 100 times) of Pearson’s correlation coefficient (R) of PSI divergence between down-sampled and original datasets.

Point 2:

I appreciate the analysis that uses additive effect sizes rather than the DeltaPSI. The number of points on the vertical in 6B is astounding to me, and I caution interpreting it as the authors do: that there is a strong asymmetric enrichment of events that have larger allelic differences in splicing after spliceosome inhibition. It is just unique to see such an asymmetric effect, so (regardless of whether intron retention events were excluded) I would like to know if there is any technical source of this asymmetric bias. Specifically, are the exons chosen to be analyzed (and more specifically the ones on the vertical) symmetrical with respect to their prevalence in treated and untreated conditions? Furthermore, rather than just showing the additive effect sizes genome-wide increase in treated (FigR7), can you show that this is really what is happening to the highlighted events (on the vertical) in Figure6B. In other words, if you plot 6B with additive effects instead of deltaPSI, are there still lots of events on the vertical? If those quality controls check out, I will be more reassured that the author’s results are not due to some technical artefact.

Answer:

In Figure 6B, we plotted all SE events that were detected both in control and PladB treated samples. Therefore, there is no bias towards either one of the samples in choosing events for the comparison. Furthermore, following the reviewer's suggestion, we plotted the divergence of the same events using additive effects instead of deltaPSI, as shown Figure R2. Since we chose the lower PSI value as the "starting PSI", all additive effects have positive values. As shown in Figure R2, the points above the dashed line (with a slope of 1) represented events with larger divergence after treatment, whereas those below the dashed line represented events with smaller divergence after treatment. In total, there are 2,764 events (77%) and only 818 points (23%) above and below the dashed line, respectively. This again demonstrated the increased allelic splicing divergence after PlaB treatment. Finally, as asked by the reviewer, to check whether the asymmetric pattern was consistent with that in Figure 6B, we labelled the points according to their classification in Figure 6B. As shown in Figure R2, the vertical points (the red points) in Figure 6B, i.e. the events divergent only in treated samples) largely remained along the vertical line, whereas those horizontal points (the blue points) in Figure 6B, i.e. the events divergent only the control samples, largely remained along the horizontal line. Together, these results suggested that the pattern we observed in Figure 6B was reproducible with additive effects.

Figure R2. Additive effects between control samples (X axis) and PlacB treated samples (Y axis). We labelled the events according to their classification in Figure 6B and colored them accordingly (please note, to make this figure clearer, we used different color sets from Figure 6B). The slope of the dashed line equals to 1, which means the points above it (n=2,764) have higher additive effects (i.e. allelic divergence) in PlacB treated samples than in control samples, and the points below (n=818) have lower additive effects in PlacB treated samples than in control samples.

December 16, 2021

RE: Life Science Alliance Manuscript #LSA-2021-01333-TR

Prof. Wei Chen
Southern University of Science and Technology
Department of Biology
1088 Xueyuan Avenue
Shenzhen, Guangdong 518005
China

Dear Dr. Chen,

Thank you for submitting your revised manuscript entitled "Mammalian splicing divergence is shaped by drift, buffering in trans and a scaling law". We would be happy to publish your paper in Life Science Alliance pending final revisions necessary to meet our formatting guidelines.

- Please upload all figure files as individual ones, including the supplementary figure files; all figure legends should only appear in the main manuscript file
- please add ORCID ID for the corresponding author-you should have received instructions on how to do so
- please add the Twitter handle of your host institute/organization as well as your own or/and one of the authors in our system
- please add your main, supplementary figure, and table legends to the main manuscript text after the references section
- please add callouts for Figures S4A-I, and S7A-E to your main manuscript text;
- please include a statement indicating approval for the mouse work, and from where that approval came

A. FINAL FILES:

B. MANUSCRIPT ORGANIZATION AND FORMATTING:

Sincerely,

December 20, 2021

RE: Life Science Alliance Manuscript #LSA-2021-01333-TRR

Prof. Wei Chen
Southern University of Science and Technology
Department of Biology
1088 Xueyuan Avenue
Shenzhen, Guangdong 518005
China

Dear Dr. Chen,

Thank you for submitting your Research Article entitled "Mammalian splicing divergence is shaped by drift, buffering in trans and a scaling law". It is a pleasure to let you know that your manuscript is now accepted for publication in Life Science Alliance. Congratulations on this interesting work.

DISTRIBUTION OF MATERIALS:

Again, congratulations on a very nice paper. I hope you found the review process to be constructive and are pleased with how the manuscript was handled editorially. We look forward to future exciting submissions from your lab.

Sincerely,
